# Socioeconomic Inequalities in Mortality among Foreign-Born and Spanish-Born in Small Areas in Cities of the Mediterranean Coast in Spain, 2009–2015

**DOI:** 10.3390/ijerph17134672

**Published:** 2020-06-29

**Authors:** Adriana Oliva-Arocas, Pamela Pereyra-Zamora, José M. Copete, Carlos Vergara-Hernández, Miguel A. Martínez-Beneito, Andreu Nolasco

**Affiliations:** 1Research Unit for the Analysis of Mortality and Health Statistics, Department of Community Nursing, Preventive Medicine, Public Health and History of Science, University of Alicante, 03080 Alicante, Spain; adriana.oliva@ua.es (A.O.-A.); copetealacant@yahoo.co.uk (J.M.C.); nolasco@ua.es (A.N.); 2Área de Desigualdades en Salud, Fundación para el Fomento de la Investigación Sanitaria y Biomédica de la Comunitat Valenciana (FISABIO), 46035 Valencia, Spain; vergara_car@gva.es; 3Departament d’Estadística i Investigació Operativa, Universitat de València, 46100 Valencia, Spain; miguel.a.martinez@uv.es

**Keywords:** mortality, socioeconomic factors, emigrants and immigrants, small-area analysis, Spain

## Abstract

Many studies have analysed socioeconomic inequalities and its association with mortality in urban areas. However, few of them have differentiated between native and immigrant populations. This study is an ecological study of mortality by overall mortality and analyses the inequalities in mortality in these populations according to the level of deprivation in small areas of large cities in the Valencian Community, from 2009 to 2015. The census tract was classified into five deprivation levels using an index based on socioeconomic indicators from the 2011 census. Rates and relative risks of death were calculated by sex, age, level of deprivation and country of birth. Poisson regression models have been used. In general, there was a higher risk of death in natives at the levels of greatest deprivation, which did not happen in immigrants. During the 2009–2015 period, there were socioeconomic inequalities in mortality, particularly in natives, who presented a higher risk of death than immigrants. Future interventions and social policies should be implemented in order to reduce inequalities in mortality amongst socioeconomic levels and to maintain the advantage that the immigrant population enjoys.

## 1. Introduction

During the last decades, research interest on the effects of the area of residence on health, taking into account individual as well as contextual factors such as socioeconomic conditions has increased [1,2]. Furthermore, important projects, both at European (INEQCITIES) and Spanish level (MEDEA) have focused on analysing socioeconomic inequalities in mortality in urban areas of a large number of cities [3,4,5]. As a result, heterogeneous patterns in these inequality trends were observed in both, Spain and Europe. In Europe, for instance, while most countries showed trends reducing socioeconomic inequalities in mortality [6,7], in others, such as Lithuania or Ireland, these increased instead [8,9]. In the case of Spain, despite the fact that mortality rates have decreased in recent years, socioeconomic inequalities in mortality have remained stable or decreased over time, although with differences according to sex, city and specific causes of mortality [10,11,12]. 

In this regard, numerous studies have used deprivation indices to highlight the relationship between the characteristics of the area of residence and risk of mortality. These indices, based on various socioeconomic indicators, have been designed in order to measure deprivation. That is, the disadvantages of an individual, a family or a group with respect to their community, or society [13]. In Spain, the worth of deprivation indices, devised within the framework of the MEDEA projects is shown in its studies on socioeconomic inequalities in mortality in urban environments [3,4,5,10,11,12,14]. In general terms, it has been found that the areas with greatest deprivation, segregation and marginalization, located in the most socioeconomically disadvantaged neighbourhoods concentrated the population with the worse health outcomes [15,16,17]. Likewise, it is well known that a large part of the immigrant population resides mainly in these urban areas [18,19]. Spain, despite its short history of immigration, has become over the course of the last 20 years one of the countries with the highest proportion of immigrants in the world. In fact, in 2008, the immigrant population represented 13.1% of the Spanish population [20]. 

After years of economic growth and job availability, the economic crisis affected Europe. Spain was one of the countries that most strictly applied severe austerity measures in social expenditure which affected the provision and access to public healthcare in general and aggravated the most vulnerable groups’ inequalities in particular [21,22]. Although many studies have shown the immigrant population in the European context in a more favourable situation in terms of mortality as compared to the native population, several authors have described the immigrant community as a highly vulnerable population in terms of health outcomes [19,23]. Other authors, nevertheless, explain how the conditions immigrants face in the host country, such as job insecurity, poor living conditions, or barriers to accessing healthcare, might be reversing these good health standards which could worsen or even disappear in a context of economic crisis [24,25]. 

In addition, several studies have also shown a great variability vis-à-vis the impact of the economic crisis on health inequalities. Some, for instance, seem to point that the economic crisis might not have propelled an increase of inequalities in Europe [26,27]. Gotsens M, et al., in this regard, has argued that socioeconomic inequalities, both at a national level or in urban areas have remained stable in Spain after the onset of the financial crisis [28]. 

Other studies carried out over the last decades have suggested that the immigrant population could influence the existence of inequalities in mortality. However, research conducted in Canada and Norway analysed inequalities separating the population according to their origin and found that these occurred independently of the migratory situation. These studies concluded that it was not due to the immigrant population but to the particular socioeconomic conditions [29,30]. 

So far, few studies have analysed inequalities in overall mortality in small areas during the economic crisis in Spain taking into account the level of deprivation and country of birth of the population. Therefore, the availability of updated socioeconomic indicators based on the 2011 Spanish Population and Housing Census is an opportunity to undertake studies of this type.

The aim of this study is to analyse the socioeconomic inequalities in mortality in native and immigrant populations in small areas of the larger cities of the Valencian Community (Alicante, Castellón and Valencia) during the period after the start of the economic crisis, since 2009 to 2015.

## 2. Materials and Methods

### 2.1. Research Design

This is an ecological study of overall mortality that analyses all deaths occurring from 2009 to 2015 in the cities of Alicante, Castellón and Valencia. These cities are located in the Southeast of Spain, on the Mediterranean coast, in the Valencian Community.

#### Data Source

All residents’ deaths in these cities during the study period were included in the analysis. Anonymized data obtained from the Valencian Community Mortality Registry were also used, as well as the variables age (0–44, 45–64, ≥65), sex (man, woman), city (Alicante, Castellón and Valencia), country of birth (Spain, other country) and cause of death. Causes of deaths were coded according to the tenth International Statistical Classification of Diseases (ICD-10) and grouped according to their large groups [31]. In addition, deaths were geo-referenced and assigned to their census tracts (CT) of residence. As this research was based on administrative data obtained retrospectively, the approval of the ethical committee is not necessary in Spain.

In every city, a deprivation index (DI) was calculated for each particular CT with the following indicators: unemployment, manual workers, temporary workers, low educational level in young people (16 to 29 years old) and low educational level in general (all them in percentage). Data were obtained from the 2011 Population and Housing Census.

These indicators had been previously proposed by the MEDEA research group for the construction of the deprivation index by means of a principal component analysis based on the census data in the main Spanish cities [32]. The deprivation index used was developed within the framework of the MEDEA3 project (third edition of the national coordinated MEDEA project) from which the study data, both socioeconomic and mortality, were obtained.

For each city, percentile 10 (P10), 25 (P25), 75 (P75) and 90 (P90) were calculated for DI, classifying CTs into five deprivation levels (DL) according to their value: DL1, values of DI less than P10; DL2, DI values between P10 and P25; DL3, DI values between P25 and P75; DL4, DI values between P75 and P90 and DL5, DI values higher than P90. Table 1 shows the number of CTs in each level and Figure 1 shows the location of CTs in each city. This classification was defined, according to the objective of the study, to preferably quantify the risks between the most socioeconomically favoured areas (DL1) and the most deprived one (DL5). Population data necessary to calculate the mortality indicators (rates and relative risks) grouped by age, sex, city and country of birth, were obtained from the statistical authority of the region, the Valencian Institute of Statistics (IVE) (Table 1).

### 2.2. Analysis Methodology

Mortality rates were calculated and plotted by sex (male, female), age group (0–44, 45–64, 65 and over), country of birth (Spain, Other country) and DL. For the estimation of the relative risks (RR) between the categories of the variables under study, Poisson regression models were adjusted, with effects of age, DL and country of birth. They were also separated by sex and a robust estimation was used to control the possible over-dispersion of data. In order to compare the mortality profile by group of cause of death according to country of birth, the proportional mortality of the large groups of the ICD-10 was calculated according to sex, country of birth and DL. For the calculation, proprietary software for calculating mortality indicators and the statistical program SPSS v.25® were also used.

## 3. Results

During the study period, a total of 78,620 deaths have occurred in the three cities under study (18,731 in Alicante, 9453 in Castellón and 50,436 in Valencia). Of these, 1049 (1.3%) could not be geo-referenced and assigned to the CT of residence because the residence address was inexistent in the registry or it did not correspond to the cities under study. Of the 77,571 deaths available for the analysis, the country of birth could not be identified in 702 (0.9%) cases, resulting in a total of 76,869 deaths for the analysis (18,330 in Alicante, 9332 in Castellón and 49,207 in Valencia).

Table 1 presents the average annual population of the three cities for the study period, stratified by age group, sex, DL and country of birth. It can be seen that the global percentage of (foreign-born) is high, 18.5% in men and 16.7% in women. When looking at the percentages according to DL, in both men and women it can be observed that in the younger age groups the percentages of the foreign population grow as the DL worsens, while in the 65 years of age and over the opposite is the case.

Table 2 shows the descriptive characteristics of the DI and of each of the five indicators used in its construction, globally and according to DL categories. As expected, all the indicators showed a range going from best to worse, from DL1 to DL5. In Table A1 and Table A2 of the Appendix A, the values of these indicators can be consulted for each of the cities studied.

Below, Table 3 shows the frequencies of death (and percentage with respect to the total) that occurred in the three cities by the large groups of the ICD-10, and by DL, sex and country of birth. As it can be seen, the three main causes of death in natives, in both men and women, are tumours, diseases of the circulatory system and diseases of the respiratory system. However, external causes are the third cause of death as regards the foreign-born, displacing diseases of the respiratory system; this is especially so in men (15.3%).

In men, the groups of causes of death such as infectious diseases, conditions originating in the perinatal period, congenital malformations, poorly defined signs and symptoms, and external causes presented higher percentages among the foreign-born than among the Spanish-born. On the contrary, tumours, endocrine and metabolic diseases, mental disorders, and diseases of the respiratory system were less abundant among foreign-born than Spanish-born.

In women there are some differences due to the fact that the groups of tumours, perinatal mortality, congenital malformations, ill-defined signs and symptoms and external causes affect immigrants in higher percentages. Nevertheless, illnesses, such as mental disorders, diseases of the nervous system, diseases of the circulatory system, diseases of the respiratory system and diseases of the genitourinary system, affect less the immigrants than the natives.

Figure 2 shows the mortality rates for overall mortality by age group, sex, country of birth and DL for the three cities studied. As the different graphs suggest the existence of possible interactions these were analysed through Poisson models with age, country of birth and DL category effects, separating by sex, and including second-order interaction terms between the three variables. Furthermore, this interaction was found to be significant (*p* < 0.001) in both men and women, suggesting a specific RR estimate for each sex, age group, and country of birth when estimating RRs between DL categories and also for each sex, age group and DL when estimating RRs among categories of country of birth.

Table 4 shows the RRs between DLs by sex, age group and country of birth. In the Spanish-born population, both in men and women, we can verify that in the younger ages (0–44 and 45–64), the DL presents significant RRs in the most depressed levels (DL5 and DL4) as compared to the most favoured one (DL1), the one used as a reference in the analysis. And it reaches a RR of about 2 in the most depressed level. In the age group of over 65 years the RRs are lower, not significantly higher than 1, for women. Regarding foreign-born, it can be seen how their behaviour is different for all ages in men, for whom RRs are predominantly less than 1 (in some cases even significantly). For women the RRs are not significant in any case either, although in the 45–64 years’ age group the RR estimates are greater than 1.

Table 5 shows the RRs of death of natives vs. immigrants, specific by age, sex and DL. In general terms Spanish-born have a higher risk of death in all situations, although this risk does not reach statistical significance neither in the DL1 for younger men and women (0–44), or in the DL2 for younger men (0–44). Besides, it can also be appreciated how the RRs grow with age, particularly in DL5.

## 4. Discussion

### 4.1. Main Findings

The results of this study confirm the existence of inequalities in general mortality in the three cities for the period 2009–2015 in relation to levels of deprivation of the area of residence, both in natives and immigrants, but with differences between these two groups. The relevance of some particular causes of death with respect to the total of deaths was different between the native and immigrant population depending on the levels of deprivation. The three cities studied showed a heterogeneous geographical distribution according to the levels of deprivation, observing a more dispersed pattern in Castellón and Valencia than in Alicante, where the most deprived areas were concentrated in the northern part of the city.

Regarding the results of the analysis of overall mortality, higher risks of death were observed in the native population with respect to the immigrant one in all DL and for all ages (except men and women of 0–44 years at the DL1, and the men of 0–44 years in DL2). Other studies have documented this immigrants’ advantage in mortality in comparison with the native population [33,34,35]. A range of studies have been carried out in order to explain this phenomenon in different countries. One of the most consistent is the one known as the “healthy immigrant effect”, in which the very act of migrating would imply having a better state of health and would maintain low levels of mortality in the host countries with respect to the native population [36]. However, some authors have also described mechanisms that could refute this explanation, since, for instance, deaths of immigrants who return to die to their home countries might be underestimated in the host country (the ‘salmon bias’). However, some authors reflect that these factors, although they could act, would still not fully explain these advantages [19,37]. 

Moreover, our results show that the immigrant population maintains this advantage regardless of the level of deprivation. That is, while the native population shows higher RRs as the socioeconomic level slopes, the RRs remained stable in immigrants. This has already been seen in studies in Canada [29] and Norway [30], the results of which showed an association between general mortality and inequalities due to socioeconomic factors (i.e. education and income) in the native population, while this was not the case in immigrants. In the city of Barcelona, a study on premature mortality conducted by Rodríguez-Sanz M et al. obtained similar results [38]. 

Another important consideration is that despite risk factors such as stress, poverty, discrimination or language barriers that might affect the immigrant population upon arrival in the host country they might also encounter protective elements to counteract them. The literature has described the existence of cultural elements through which the immigrant population would keep more favourable mortality results due to a healthier lifestyle habits (consumption of tobacco, alcohol or diet) that they would have brought with them from their countries of origin [39,40]. Furthermore, the formation of social support networks at the community level, or family ties, in the receiving countries could also act as a cushion against the effects of low socioeconomic conditions on health [41]. In another direction, mechanisms related to the duration of the immigrant’s stay in the host country that could mitigate the effects of the mortality advantage that the effect of the healthy immigrant provides have also been described. It has been shown that as the immigrant remains in the host country, assimilation and adaptation to local lifestyles would make mortality risks to converge towards similar levels. This would mean losing their mortality advantage [42]. 

The results of this study show that middle-aged women (45–64 years) are the only ones who present some inequality by levels of deprivation (although not significant). According to gender, we found that studies such as that of Oksuzyan A. et al. [43] seem to grant some advantage in excess mortality to the female population, however, in others such as that of Boulogne R et al. [34] women seemed to be at a disadvantage. Regarding age, Guillot M. et al. [37] obtained consistent results between different countries and immigrants on a possible U-shaped mortality pattern, in which at early ages they would have a higher risk of mortality, to later gain advantage at intermediate ages and finally converge in old age in risk levels similar to that of the natives. Although in this study it was not possible to disaggregate the results by the country of origin of the immigrant population, multiple investigations seem to coincide in a lower mortality for those from non-western countries. However, immigrants from countries of Eastern Europe or Africa (North of Africa or sub-Saharan Africa) could be especially vulnerable, presenting higher risks of death [34,35,43,44]. Finally, studies such as that of Syse A. et al. and Aldridge RW. et al [23,33] indicate that this mortality advantage could also be shared by other type of immigrants such as refugees or those who migrate for family reasons, but not for asylum seekers.

When describing mortality according to major groups of diseases it was found that despite the fact that the main causes of death in natives and immigrants show similar proportional mortality patterns, there are important inequalities in their magnitude according to sex and causes of death. Hence, regarding the excess proportional mortality from infectious and parasitic diseases, our results are consistent with those observed in previous studies [33,44]. This means that despite the differences observed in proportional mortality from this cause, its relevance in the general mortality of the immigrant population appears to be low. In this regard, various authors seem to indicate that the incidence of infectious and parasitic diseases comes mainly from the countries of origin [45] and, despite the limitations presented by studies with an ecological design to establish causal relationships, low mortality could also be related to adequate access to the national health system and treatments.

It is important to highlight that in our analysis it was possible to observe how the proportional mortality due to external causes in the immigrant population, in both sexes, maintained high frequencies throughout all levels of deprivation and greater impact, as compared to the native. There is evidence that the immigrant population suffers more work-related diseases and injuries than the native population due to the performance of unskilled jobs, in areas such as construction, agriculture and transportation, which carry risks and lack protection measures [46]. 

Regarding the differences in proportional mortality due to conditions originating in the perinatal period, a study in the same cities of our research by Barona-Villar et al. observed an excess risk in the immigrant population in comparison to the native population, especially in those from Eastern Europe and Sub-Saharan Africa, and a risk of more than double perinatal mortality caused by late infectious diseases [47]. 

Finally, the results obtained from the analysis of proportional mortality from tumours should be highlighted. Immigrant men showed a favourable pattern of proportional mortality with respect to the natives through all levels of deprivation. However, in the case of women the situation was the opposite, placing them by far at a clear disadvantage in DL4 (23.0% in natives versus 48.2% in immigrants). These results seem to be contrary to those observed in various studies in the case of women [34,35,44]. Despite the limitations to establish causal relationships, according to the literature, it could be pointed out that the excess mortality of immigrant women could be due to a low use of the screening program for some groups of immigrant women in Spain [48]. Future analyses of cancer mortality could also shed light on these results.

### 4.2. Methodological Strengths and Limitations

It should be borne in mind that this is an ecological study, with the limitations of this type of study. Thus, the results obtained do not allow to infer a causal association and the relationship obtained between the DL and the risks of death when using the CTs may not be applicable at the individual level (ecological fallacy), reflecting both the effect of the individual socioeconomic level and the contextual effect of the residence area.

As in any study of the effect of the area of residence, it must be taken into account that exposure to the risks of death for some causes may have occurred in places other than the place of residence, for example, at work. Thus, those who live in more depressed neighbourhoods could also be the most exposed at risk.

The data analysis has been carried out jointly for the 3 cities. This is mainly due to reasons of statistical power. However, if the descriptive characteristics of the socioeconomic indicators of the cities are observed (Appendix A) we see that they do not present great differences among the cities, with a behaviour consistent with the different deprivation levels. In addition, analyses on mortality were carried out to establish the existence of significant interactions between the city and the DL, finding these results not significant. Hence, it cannot be stated that the association between DL and mortality is different by city.

Another limitation comes from difficulties in geo-referencing the totality of deaths, but the percentage of not georeferenced death was very small (1.3%), lower than usual in this type of study. It should also be borne in mind that some deaths could not be included in the analyses as country of birth was not available in the registry. Their percentage, nevertheless, was also very small (0.9%). These shortcomings should have little effect on the results obtained.

The deprivation index was obtained from indicators from the 2011 census, a year that is located approximately in the centre of the period studied. Furthermore, no significant changes at the level of the census tract throughout the study period are expected. This particular index is not the only option, but it was chosen for this research because it had already been successfully applied in most of the previous studies on mortality inequalities in cities in Spain, and therefore comparison with other research was possible. Similarly, its classification in different levels of deprivation is not the only choice either, but it responds well to the objective of evaluating the inequality between the population with the highest and lowest levels of deprivation, with consistent results across the different categories used. On the other hand, information on other lifestyle variables such as tobacco or alcohol consumption was not available.

In relation to proportional mortality, it must be noticed that only deaths are taken into account in its calculation. In some cases, it could be affected by the youth of immigrants as compared to natives.

In this research it was not possible to disaggregate by the specific country or region of birth, and although it would have been desirable to separate at least by regions of economic and non-economic immigration it was not possible due to the preservation of statistical secrecy in small areas. Understandingly, there are limitations in access to information on populations, data from the Mortality Registry and the Population Census. Future research should further develop this aspect.

A last consideration in relation to the immigrant population is that Spanish data sources do not include immigrants in a situation of illegal residence in Spain. So, it is necessary to enable mechanisms that allow the inclusion of this population for all purposes, particularly those related with health, since it is probable that these undocumented groups are suffering of greater vulnerability.

## 5. Conclusions

This study shows the existence of socioeconomic inequalities in mortality in the larger cities of the Valencian Community, both in the native and immigrant population, during the period 2009–2015. These inequalities are lower for the immigrant population and, at the same age groups, immigrants also present lower risks of death than the native population at all levels of deprivation, both for men and women. The analysis identified that in some of the large groups of diseases the proportional mortality is higher in the immigrant population at all levels of deprivation than in the native population. Finally, this study has identified the areas and populations at greatest risk on which to implement social interventions and health policies aimed at reducing existing socioeconomic inequalities among population groups, particularly in the native population. Future interventions and social policies should be implemented in order to reduce inequalities in mortality amongst socioeconomic levels and to maintain the advantage that the immigrant population enjoys.

## Figures and Tables

**Figure 1 ijerph-17-04672-f001:**
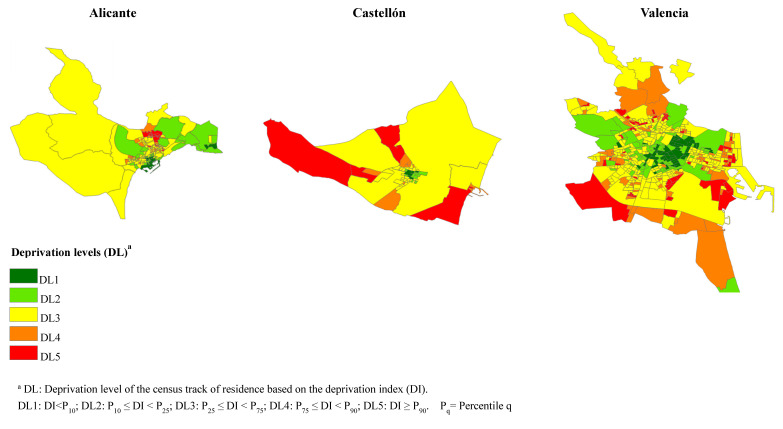
Geographical distribution of the five levels of deprivation (DL) according to census tracts in the cities of Alicante, Castellón and Valencia—2011.

**Figure 2 ijerph-17-04672-f002:**
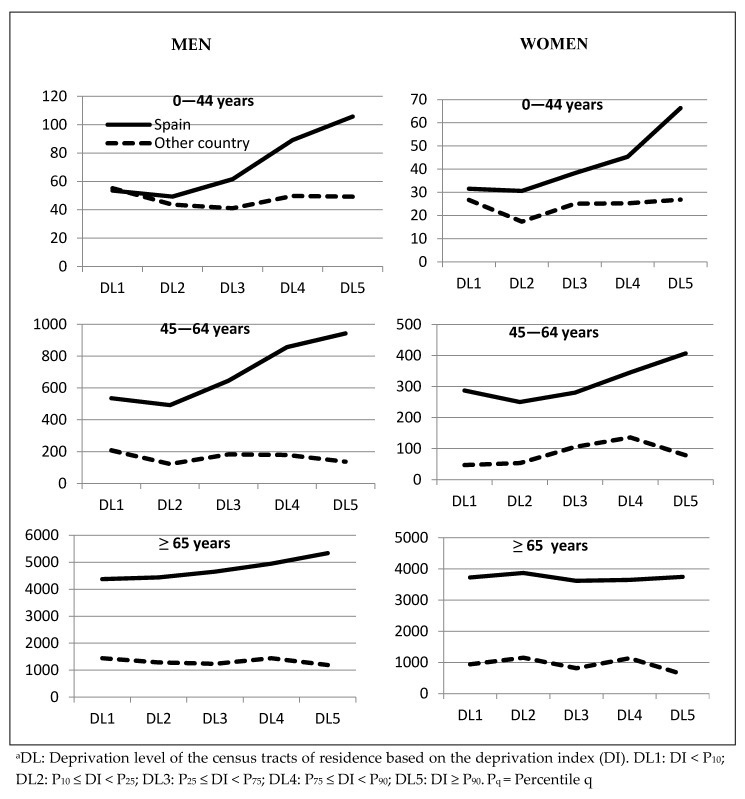
Mortality rates per 100,000 inhabitants by sex, age groups, country of birth and level of deprivation (DL) ^a^.

**Table 1 ijerph-17-04672-t001:** Average annual population for the three cities by age group, sex and level of deprivation according to census tracts and country of birth between the years 2009 to 2015.

Age	Deprivation Level (DL) ^a^	Men			Women		
		Native	Foreign Born	Foreign Born %	Native	Foreign Born	Foreign Born %
0–44	DL1	18,666	4140	18.2	18,590	4272	18.7
	DL2	57,395	11,774	17.0	56,494	12,343	17.9
	DL3	151,929	44,809	22.8	147,412	42,714	22.5
	DL4	36,188	14,912	29.2	34,647	13,003	27.3
	DL5	23,503	12,480	34.7	21,744	9572	30.6
45–64	DL1	10,002	1372	12.1	12,198	1514	11.0
	DL2	25,926	3937	13.2	29,334	4251	12.7
	DL3	73,686	12,662	14.7	81,954	13,407	14.1
	DL4	17,535	3885	18.1	18,978	3686	16.3
	DL5	10,103	3006	22.9	10,534	2537	19.4
≥65	DL1	7334	346	4.5	11,606	531	4.4
	DL2	14,590	1031	6.6	21,660	1310	5.7
	DL3	46,483	1913	4.0	68,298	2694	3.8
	DL4	12,894	415	3.1	19,209	653	3.3
	DL5	8229	299	3.5	12,088	481	3.8
**Total**		**514,464**	**116,982**	**18.5**	**564,745**	**112,968**	**16.7**

**^a^** DL: Deprivation level of the census tracts of residence based on the deprivation index (DI). DL1: DI < P_10_; DL2: P_10_ ≤ DI < P_25_; DL3: P_25_ ≤ DI < P_75_; DL4: P_75_ ≤ DI < P_90_; DL5: DI ≥ P_90_; P_q_ = Percentile q.

**Table 2 ijerph-17-04672-t002:** Descriptive characteristics of the Deprivation Index and socioeconomic indicators according to deprivation levels.

Deprivation Index or Socioeconomic Indicator	Deprivation Level (DL) ^a^	Number of CTs	Mean	Standard Deviation	Minimum	Maximum
Deprivation Index	DL1	75	−0.736	0.098	−1.027	−0.485
	DL2	115	−0.502	0.092	−0.715	−0.337
	DL3	386	−0.009	0.192	−0.417	0.319
	DL4	116	0.437	0.082	0.307	0.601
	DL5	75	0.877	0.276	0.494	2.255
	Total	767	0.000	0.474	−1.027	2.255
Manual workers	DL1	75	0.159	0.047	0.074	0.301
	DL2	115	0.256	0.060	0.158	0.440
	DL3	386	0.470	0.100	0.237	0.724
	DL4	116	0.633	0.069	0.415	0.806
	DL5	75	0.718	0.075	0.553	0.881
	Total	767	0.456	0.182	0.074	0.881
Unemployed	DL1	75	0.213	0.039	0.157	0.332
	DL2	115	0.246	0.039	0.160	0.352
	DL3	386	0.304	0.052	0.196	0.469
	DL4	116	0.354	0.052	0.196	0.485
	DL5	75	0.412	0.076	0.265	0.655
	Total	767	0.305	0.074	0.157	0.655
Temporary workers	DL1	75	0.116	0.028	0.051	0.181
	DL2	115	0.138	0.035	0.042	0.224
	DL3	386	0.175	0.044	0.072	0.416
	DL4	116	0.218	0.055	0.081	0.375
	DL5	75	0.250	0.066	0.141	0.460
	Total	767	0.178	0.059	0.042	0.460
Low educational level	DL1	75	0.082	0.025	0.034	0.143
	DL2	115	0.124	0.027	0.071	0.248
	DL3	386	0.204	0.045	0.094	0.340
	DL4	116	0.283	0.038	0.196	0.389
	DL5	75	0.367	0.066	0.248	0.695
	Total	767	0.208	0.088	0.034	0.695
Low educational level in young people (16 to 29 years old)	DL1	75	0.021	0.015	0.007	0.106
	DL2	115	0.035	0.020	0.003	0.099
	DL3	386	0.065	0.034	0.012	0.185
	DL4	116	0.112	0.049	0.028	0.299
	DL5	75	0.220	0.107	0.074	0.772
	Total	767	0.078	0.070	0.003	0.772

^a^ DL: Deprivation level of the census tracts of residence based on the deprivation index (DI). DL1: DI < P_10_; DL2: P_10_ ≤ DI < P_25_; DL3: P_25_ ≤ DI < P_75_; DL4: P_75_ ≤ DI < P_90_; DL5: DI ≥ P_90_; P_q_ = Percentile q.

**Table 3 ijerph-17-04672-t003:** Frequencies and percentages of death according to large groups diseases of the ICD-10, by sex, level of deprivation and country of birth, for the three cities. 2009–2015.

MEN	DEPRIVATION INDEX (DI)												TOTAL
	DL1: DI < P_10_		DL2: P_10_ ≤ DI < P_25_		DL3: P_25_ ≤ DI < P_75_		DL4: P_75_ ≤ DI < P_90_		DL5: DI ≥ P_90_		Total		
ICD-10 GROUP	Spain	Other Country	Spain	Other Country	Spain	Other Country	Spain	Other Country	Spain	Other Country	Spain	Other Country	
I Certain infectious and parasitic diseases	25	3	95	4	322	14	106	9	96	3	644	33	677
	0.9%	4.2%	1.7%	2.4%	1.7%	3.1%	1.8%	6.3%	2.5%	3.0%	1.7%	3.5%	1.8%
II Neoplasms	872	21	1936	52	6665	127	1919	42	1327	26	12719	268	12987
	32.4%	29.6%	34.4%	31.7%	34.9%	27.7%	33.4%	29.2%	33.9%	26.3%	34.3%	28.6%	34.1%
III Diseases of the blood and blood-forming organs and immunity	10	0	20	1	51	1	17	1	10	0	108	3	111
	0.4%	0.0%	0.4%	0.6%	0.3%	0.2%	0.3%	0.7%	0.3%	0.0%	0.3%	0.3%	0.3%
IV Endocrine, nutritional and metabolic diseases	61	0	128	3	446	3	158	1	103	2	896	9	905
	2.3%	0.0%	2.3%	1.8%	2.3%	0.7%	2.8%	0.7%	2.6%	2.0%	2.4%	1.0%	2.4%
V Mental and behavioural disorders	73	0	160	3	581	6	167	2	114	1	1095	12	1107
	2.7%	0.0%	2.8%	1.8%	3.0%	1.3%	2.9%	1.4%	2.9%	1.0%	3.0%	1.3%	2.9%
VI–VIII Diseases of the nervous system and the organs of the senses	138	3	329	5	932	18	249	2	165	2	1813	30	1843
	5.1%	4.2%	5.9%	3.0%	4.9%	3.9%	4.3%	1.4%	4.2%	2.0%	4.9%	3.2%	4.8%
IX Diseases of the circulatory system	826	21	1593	48	5333	134	1592	35	1032	30	10376	268	10644
	30.7%	29.6%	28.3%	29.3%	27.9%	29.3%	27.7%	24.3%	26.4%	30.3%	28.0%	28.6%	28.0%
X Diseases of the respiratory system	350	3	649	13	2326	24	724	11	529	10	4578	61	4639
	13.0%	4.2%	11.5%	7.9%	12.2%	5.2%	12.6%	7.6%	13.5%	10.1%	12.3%	6.5%	12.2%
XI Diseases of the digestive system	122	6	239	11	925	20	311	10	212	2	1809	49	1858
	4.5%	8.5%	4.3%	6.7%	4.8%	4.4%	5.4%	6.9%	5.4%	2.0%	4.9%	5.2%	4.9%
XII Diseases of the skin and subcutaneous tissue	6	0	8	0	42	1	18	0	8	0	82	1	83
	0.2%	0.0%	0.1%	0.0%	0.2%	0.2%	0.3%	0.0%	0.2%	0.0%	0.2%	0.1%	0.2%
XIII Diseases of the musculoskeletal system and connective tissue	15	0	26	0	72	2	26	1	20	1	159	4	163
	0.6%	0.0%	0.5%	0.0%	0.4%	0.4%	0.5%	0.7%	0.5%	1.0%	0.4%	0.4%	0.4%
XIV: Diseases of the genitourinary system	90	0	169	1	490	11	167	1	88	1	1004	14	1018
	3.3%	0.0%	3.0%	0.6%	2.6%	2.4%	2.9%	0.7%	2.2%	1.0%	2.7%	1.5%	2.7%
XVI Certain conditions originating in the perinatal period	3	0	14	1	32	5	10	3	8	1	67	10	77
	0.1%	0.0%	0.2%	0.6%	0.2%	1.1%	0.2%	2.1%	0.2%	1.0%	0.2%	1.1%	0.2%
XVII Congenital malformations	2	2	14	0	33	3	14	1	6	3	69	9	78
	0.1%	2.8%	0.2%	0.0%	0.2%	0.7%	0.2%	0.7%	0.2%	3.0%	0.2%	1.0%	0.2%
XVIII Symptoms, signs, not elsewhere classified	19	0	53	5	180	13	42	3	42	1	336	22	358
	0.7%	0.0%	0.9%	3.0%	0.9%	2.8%	0.7%	2.1%	1.1%	1.0%	0.9%	2.4%	0.9%
XX External causes of morbidity and mortality	79	12	190	17	693	76	223	22	155	16	1340	143	1483
	2.9%	16.9%	3.4%	10.4%	3.6%	16.6%	3.9%	15.3%	4.0%	16.2%	3.6%	15.3%	3.9%
**Total**	**2691**	**71**	**5623**	**164**	**19123**	**458**	**5743**	**144**	**3915**	**99**	**37095**	**936**	**38031**
	**100.0%**	**100.0%**	**100.0%**	**100.0%**	**100.0%**	**100.0%**	**100.0%**	**100.0%**	**100.0%**	**100.0%**	**100.00%**	**100.00%**	**100.00%**
**WOMEN**	**DEPRIVATION INDEX (DI)**												**TOTAL**
	**DL1: DI < P_10_**		**DL2: P_10_ ≤ DI < P_25_**		**DL3: P_25_ ≤ DI < P_75_**		**DL4: P_75_ ≤ DI < P_90_**		**DL5: DI ≥ P_90_**		**Total**		
**ICD-10 GROUP**	**Spain**	**Other Country**	**Spain**	**Other Country**	**Spain**	**Other Country**	**Spain**	**Other Country**	**Spain**	**Other Country**	**Spain**	**Other Country**	
I Certain infectious and parasitic diseases	54	1	98	5	327	6	80	0	97	2	656	14	670
	1.6%	2.0%	1.5%	3.6%	1.7%	1.8%	1.5%	0.0%	2.7%	3.6%	1.7%	2.1%	1.7%
II Neoplasms	715	15	1537	36	4431	124	1259	53	809	17	8751	245	8996
	21.6%	30.0%	23.6%	26.3%	23.0%	37.8%	23.0%	48.2%	22.7%	30.9%	22.9%	36.0%	23.2%
III Diseases of the blood and blood-forming organs and immunity	16	0	32	2	79	3	28	1	22	0	177	6	183
	0.5%	0.0%	0.5%	1.5%	0.4%	0.9%	0.5%	0.9%	0.6%	0.0%	0.5%	0.9%	0.5%
IV Endocrine, nutritional and metabolic diseases	88	3	196	2	680	7	199	0	144	3	1307	15	1322
	2.7%	6.0%	3.0%	1.5%	3.5%	2.1%	3.6%	0.0%	4.0%	5.5%	3.4%	2.2%	3.4%
V Mental and behavioural disorders	195	1	429	6	1104	7	325	1	228	1	2281	16	2297
	5.9%	2.0%	6.6%	4.4%	5.7%	2.1%	5.9%	0.9%	6.4%	1.8%	6.0%	2.4%	5.9%
VI–VIII Diseases of the nervous system and the organs of the senses	281	2	568	12	1578	15	431	7	281	1	3139	37	3176
	8.5%	4.0%	8.7%	8.8%	8.2%	4.6%	7.9%	6.4%	7.9%	1.8%	8.2%	5.4%	8.2%
IX Diseases of the circulatory system	1220	16	2187	43	6701	82	1896	20	1247	19	13251	180	13431
	36.8%	32.0%	33.6%	31.4%	34.7%	25.0%	34.7%	18.2%	34.9%	34.5%	34.7%	26.5%	34.6%
X Diseases of the respiratory system	312	2	605	10	1788	16	491	3	315	2	3511	33	3544
	9.4%	4.0%	9.3%	7.3%	9.3%	4.9%	9.0%	2.7%	8.8%	3.6%	9.2%	4.9%	9.1%
XI Diseases of the digestive system	122	2	242	1	905	18	276	5	154	2	1699	28	1727
	3.7%	4.0%	3.7%	0.7%	4.7%	5.5%	5.0%	4.5%	4.3%	3.6%	4.5%	4.1%	4.4%
XII Diseases of the skin and subcutaneous tissue	18	0	26	3	107	0	36	0	11	0	198	3	201
	0.5%	0.0%	0.4%	2.2%	0.6%	0.0%	0.7%	0.0%	0.3%	0.0%	0.5%	0.4%	0.5%
XIII Diseases of the musculoskeletal system and connective tissue	32	0	64	2	181	4	54	4	29	0	360	10	370
	1.0%	0.0%	1.0%	1.5%	0.9%	1.2%	1.0%	3.6%	0.8%	0.0%	0.9%	1.5%	1.0%
XIV Diseases of the genitourinary system	118	0	239	4	717	3	195	1	126	0	1395	8	1403
	3.6%	0.0%	3.7%	2.9%	3.7%	0.9%	3.6%	0.9%	3.5%	0.0%	3.7%	1.2%	3.6%
XV Pregnancy, childbirth and the puerperium	0	0	2	0	2	1	1	0	0	0	5	1	6
	0.0%	0.0%	0.0%	0.0%	0.0%	0.3%	0.0%	0.0%	0.0%	0.0%	0.0%	0.1%	0.0%
XVI Certain conditions originating in the perinatal period	3	0	13	1	21	2	11	3	5	1	53	7	60
	0.1%	0.0%	0.2%	0.7%	0.1%	0.6%	0.2%	2.7%	0.1%	1.8%	0.1%	1.0%	0.2%
XVII Congenital malformations	4	0	12	1	32	3	14	0	2	2	64	6	70
	0.1%	0.0%	0.2%	0.7%	0.2%	0.9%	0.3%	0.0%	0.1%	3.6%	0.2%	0.9%	0.2%
XVIII Symptoms, signs not elsewhere classified	66	4	104	4	199	8	50	4	36	1	455	21	476
	2.0%	8.0%	1.6%	2.9%	1.0%	2.4%	0.9%	3.6%	1.0%	1.8%	1.2%	3.1%	1.2%
XX External causes of morbidity and mortality	69	4	147	5	451	29	125	8	64	4	856	50	906
	2.1%	8.0%	2.3%	3.6%	2.3%	8.8%	2.3%	7.3%	1.8%	7.3%	2.2%	7.4%	2.3%
**Total**	**3313**	**50**	**6501**	**137**	**19303**	**328**	**5471**	**110**	**3570**	**55**	**38158**	**680**	**38838**
	**100.0%**	**100.0%**	**100.0%**	**100.0%**	**100.0%**	**100.0%**	**100.0%**	**100.0%**	**100.0%**	**100.0%**	**100,0%**	**100,0%**	**100,0%**

**Table 4 ijerph-17-04672-t004:** Relative risks of death from all causes according to level of deprivation and 95% confidence intervals (95% CI), specific for age, sex and country of birth.

Country of Birth	Age	Deprivation Level (DL) ^a^	Men			Women		
			RR	95% CI		RR	95% CI	
				Lower	Upper		Lower	Upper
Spain	0–44	DL5	1.974	1.365	2.854	2.106	1.564	2.837
		DL4	1.665	1.209	2.294	1.439	1.037	1.998
		DL3	1.150	0.831	1.589	1.215	0.904	1.632
		DL2	0.920	0.664	1.274	0.971	0.668	1.411
		DL1	1			1		
	45–64	DL5	1.761	1.683	1.843	1.418	1.225	1.641
		DL4	1.600	1.469	1.743	1.204	1.076	1.347
		DL3	1.205	1.161	1.251	0.976	0.870	1.095
		DL2	0.919	0.873	0.967	0.871	0.782	0.969
		DL1	1			1		
	≥65	DL5	1.220	1.180	1.261	1.005	0.956	1.057
		DL4	1.131	1.082	1.182	0.978	0.930	1.030
		DL3	1.064	1.040	1.087	0.971	0.929	1.015
		DL2	1.014	0.989	1.040	1.039	0.975	1.107
		DL1	1			1		
Other country	0–44	DL5	0.892	0.753	1.056	1.004	0.792	1.273
		DL4	0.902	0.674	1.208	0.945	0.692	1.289
		DL3	0.745	0.691	0.803	0.938	0.791	1.112
		DL2	0.791	0.512	1.223	0.649	0.510	0.827
		DL1	1			1		
	45–64	DL5	0.662	0.466	0.940	1.672	0.326	8.568
		DL4	0.865	0.639	1.172	2.876	0.876	9.445
		DL3	0.883	0.651	1.198	2.237	0.639	7.825
		DL2	0.593	0.425	0.826	1.140	0.285	4.566
		DL1	1			1		
	≥65	DL5	0.826	0.468	1.460	0.662	0.373	1.175
		DL4	1.001	0.642	1.561	1.207	0.658	2.216
		DL3	0.859	0.567	1.300	0.867	0.478	1.570
		DL2	0.893	0.603	1.321	1.227	0.659	2.284
		DL1	1			1		

^a^ DL: Deprivation level of the census tracts of residence based on the deprivation index (DI). DL1: DI < P_10_; DL2: P_10_ ≤ DI < P_25_; DL3: P_25_ ≤ DI < P_75_; DL4: P_75_ ≤ DI < P_90_; DL5: DI ≥ P_90_ P_q_ = Percentile q.

**Table 5 ijerph-17-04672-t005:** Relative risks of death from all causes in Spanish-born versus foreign-born (and 95% confidence intervals, 95% CI), specific for age, sex and level of deprivation.

Deprivation Level (DL) ^a^	Age	Men			Women		
		RR	95% CI		RR	95% CI	
			Lower	Upper		Lower	Upper
DL1	0–44	0.970	0.701	1.343	1.178	0.873	1.590
	45–64	2.573	1.969	3.362	6.084	1.859	19.904
	≥65	3.030	2.047	4.484	3.954	2.253	6.940
DL2	0–44	1.128	0.729	1.746	1.762	1.269	2.448
	45–64	3.989	3.256	4.886	4.646	2.241	9.634
	≥65	3.443	3.383	3.503	3.348	2.551	4.393
DL3	0–44	1.498	1.395	1.608	1.526	1.298	1.794
	45–64	3.510	3.016	4.084	2.656	1.744	4.043
	≥65	3.753	3.272	4.306	4.431	3.642	5.390
DL4	0–44	1.791	1.345	2.384	1.795	1.280	2.517
	45–64	4.757	4.029	5.616	2.547	2.197	2.953
	≥65	3.422	2.764	4.237	3.204	2.536	4.048
DL5	0–44	2.149	1.685	2.740	2.470	1.954	3.123
	45–64	6.843	5.431	8.622	5.160	1.659	16.044
	≥65	4.472	2.956	6.764	6.003	5.318	6.777

^a^ DL: Deprivation level of the census tracts of residence based on the deprivation index (DI). DL1: DI < P_10_; DL2: P_10_ ≤ DI < P_25_; DL3: P_25_ ≤ DI < P_75_; DL4: P_75_ ≤ DI < P_90_; DL5: DI ≥ P_90_ P_q_ = Percentile q.

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
