# Peer review of "Socioeconomic Inequalities in Mortality among Foreign-Born and Spanish-Born in Small Areas in Cities of the Mediterranean Coast in Spain, 2009–2015"

_ijerph, 2020, doi:10.3390/ijerph17134672_

Round 1

Reviewer 1 Report

This study discussed the inequalities in mortality in large cities in Spain, especially between the native and immigrant populations. The study was well designed and with good implication. The manuscript can be accepted in its current form. My suggestion is that the authors may provide with stronger evidence and have more interesting findings if they could get more information on healthy lifestyles, healthcare resources and its utilization in these areas, and link the current results with these variables.

Author Response

Reviewer 1

Comments and Suggestions for Authors

This study discussed the inequalities in mortality in large cities in Spain, especially between the native and immigrant populations. The study was well designed and with good implication. The manuscript can be accepted in its current form. My suggestion is that the authors may provide with stronger evidence and have more interesting findings if they could get more information on healthy lifestyles, healthcare resources and its utilization in these areas, and link the current results with these variables.

-Thank you very much for your comments. We appreciate your recommendation and we hope to incorporate it in future studies.

Reviewer 2 Report

Line 17 – First letter of the second sentence should be capitalized.

Line 37 – The word ‘Important’ should start with a lower-case letter.

Line 47 – A definition or explanation of what deprivation entails would be helpful for the reader.

Lines 71-72 – Check wording on “found that these found that…”.

Lines 153-154 – What constitutes external causes of death among the foreign-born?

Line 22 – Check for juxtaposed text at the beginning of the sentence.

Line 195 – For participants from other countries of birth, is there further information as to what regions of the world that these other countries represent?

Lines 247-249 – In light of the literature on effects of assimilation, was length of stay of immigrants within the host country (Spain) factored into the analysis of data for this study?  It will be helpful to know what countries these participants represent to help contextualize the information.

Lines 302-305 – When considering area of residence, did your study also take into account the effect of geographic location (e.g., rural versus urban)?

Line 330 – Indeed, information on specific country of birth would have been insightful for this study.  Is information available on region of birth (e.g., Eastern Europe, North Africa, Sub-Saharan Africa)?

Overall, this is an interesting and insightful paper.  With edits to improve clarity, language and style, the paper can be improved. 

Author Response

Reviewer 2

Comments and Suggestions for Authors

Line 17 – First letter of the second sentence should be capitalized.

-The suggested change has been introduced in line 18

Line 37 – The word ‘Important’ should start with a lower-case letter.

-The suggested change has been introduced on line 37.

Line 47 – A definition or explanation of what deprivation entails would be helpful for the reader.

-A description of the concept of deprivation has been added on line 46-50.

Lines 71-72 – Check wording on “found that these found that…”.

-The repeated word has been deleted on line 76.

Lines 153-154 – What constitutes external causes of death among the foreign-born?

-Chapter XX "External causes of morbidity and mortality" described by the ICD-10 includes a multitude of codes that fall under 4 major groups of causes: transport accidents, falls and various accidents, other external causes of mortality and side effects of treatments. In the references section, the reference corresponding to the ICD-10 manual is included (reference 31) in data sources line 101.

Line 22 – Check for juxtaposed text at the beginning of the sentence.

- The suggested change has been introduced on line 22.

Line 195 – For participants from other countries of birth, is there further information as to what regions of the world that these other countries represent?

-The reasons why we could not include that information are explained in lines 340-344, section 4.2 (Methodological strengths and limitations). The phrase in line 340 has been modified for clarification.

Lines 247-249 – In light of the literature on effects of assimilation, was length of stay of immigrants within the host country (Spain) factored into the analysis of data for this study?  It will be helpful to know what countries these participants represent to help contextualize the information.

-Unfortunately, this information is not available in Spain as it is not included in mortality records.

Lines 302-305 – When considering area of residence, did your study also take into account the effect of geographic location (e.g., rural versus urban)?

-As the analysis is oriented towards urban inequalities this effect was not contemplated.

Line 330 – Indeed, information on specific country of birth would have been insightful for this study.  Is information available on region of birth (e.g., Eastern Europe, North Africa, Sub-Saharan Africa)?

-The authors appreciate your comment. Unfortunately, it was not possible to obtain desegregated data by specific country or region of birth. In the line 343 it has been specified that it was not possible to obtain information on region of birth.

Overall, this is an interesting and insightful paper.  With edits to improve clarity, language and style, the paper can be improved. 

Thank you very much for your comments. They have been of great help for the edition and to improve the clarity of the article.

Reviewer 3 Report

Comments to the Author
The authors use Valencian community data in Spain to show that relative socioeconomic status (SES) in equality and mortality both in the natives and immigrant population. While there is much to recommend about this paper I have a couple of concerns.

 Major comments:

  1. Please provide references for using deprivation index as a measure for SES inequality. Is this valid? And have others done this before?
  2. How stable are the socioeconomic deprivation under consideration? Was there any information available about the amount of time participants resided at the coded addresses, or if they moved at some point across the study period?
  3. Interaction between SES and individual characteristics is need. If they were significant, I would suggest that you report them and explain in more detail why they were not meaningful. In any case, it is important to discuss interaction effects (or the absence thereof) in the discussion.
  4. The authors recognize several limitations, but not sufficiently low number of included outcomes of mortality from some counties that contribute to the greatest statistical significance. Therefore, I would suggest taking into account this limitation in sentences of results and discussion section.
  5. One of the limitations is that the data does not include information on traditional risk factors including smoking, excessive alcohol drinking and/or comorbidities that may be related to the risk of mortality. This should be, however, included in the discussion as a limitation.
  6. The authors have not controlled for smoking, which may be more easily done at the county level using smoking rates than at the individual level if it is not measured. The reason I wonder about the role of smoking is that I think it may be related to mortality since it is so strongly linked to SES. If individuals living in areas of low level of SES are also either more likely to smoke or are more likely to have neighbors, friends, and family who smoke then you may have one mechanism through which social networks impact mortality.
  7. The results suggest an independent effect for mortality of county-level context, however, the analysis lack individual level data. Individual-level SES effects could be more apparent than aggregate SES effects or at the very least that they would provide independent or confounded impacts on mortality; this should be, however, included in the discussion.
  8. Please have this paper edited by a proficient, native English speaker or, preferably, by one of the many science editing services now available.

Minor comments

Line 17. The sentence is incomplete, it should be “many studies have analyzed socioeconomic inequalities and association with mortality in urban areas”.

Line 22, please clarify “specific rates”.

Line 24-26, two sentences were repeated stated.

Line 40-44, please clarify the changing level of inequality regards for socioeconomic inequality or inequality mortality.

Line 50-53, please revise the sentence

Line 57-60, please revise the sentence

Line 78-81, please clarify the aim of the study.

Line 103, please clarify “come”

Line 111-112, please edit the sentence, and clarify “mortality indicators”

Line 115, “specific mortality”. The present study only shown rates and RR for overall mortality.

Line 171 and Line 177, Figure legend: “specific mortality rates”, it should be over mortality rates by age, gender, country of birth, and DL.

Line 202, please clarify “DL5.ble 5.

Line 202-203, the sentence is not fully stated.

Line 343, please clarify the inequality regards for socioeconomic inequality or inequality mortality.

Author Response

Reviewer 3

Comments to the Author
The authors use Valencian community data in Spain to show that relative socioeconomic status (SES) in equality and mortality both in the natives and immigrant population. While there is much to recommend about this paper I have a couple of concerns.

 Major comments:

  1. Please provide references for using deprivation index as a measure for SES inequality. Is this valid? And have others done this before?

-From lines 46 to 50, in the introduction, we have included a paragraph expanding the information on the use of deprivation indices in the context of the study of socioeconomic inequalities in mortality.

  1. How stable are the socioeconomic deprivation under consideration? Was there any information available about the amount of time participants resided at the coded addresses, or if they moved at some point across the study period?

-This index has been used in Spain in numerous studies in the framework of the Medea project, as mentioned in the text. Its effectiveness for measuring deprivation and detecting socioeconomic inequalities in small areas has been shown in several studies. Unfortunately, no information was available on the amount of time participants resided at the coded addresses, or if they moved at some point across the study period.

  1. Interaction between SES and individual characteristics is need. If they were significant, I would suggest that you report them and explain in more detail why they were not meaningful. In any case, it is important to discuss interaction effects (or the absence thereof) in the discussion.

-Individual characteristics have not been incorporated because this is an ecological study. Instead, a specific analysis by age groups, sex, level of deprivation and country of birth has been carried out.

  1. The authors recognize several limitations, but not sufficiently low number of included outcomes of mortality from some counties that contribute to the greatest statistical significance. Therefore, I would suggest taking into account this limitation in sentences of results and discussion section.

- Association analyses were performed with the frequency and population of each age group, sex, level of deprivation and country of birth. In addition, we have mentioned significance in parts of the discussion

  1. One of the limitations is that the data does not include information on traditional risk factors including smoking, excessive alcohol drinking and/or comorbidities that may be related to the risk of mortality. This should be, however, included in the discussion as a limitation.

We appreciate your comments. We have added a limitation in lines 335-336 stating that information on traditional risk factors, including smoking and excessive alcohol was not available.

  1. The authors have not controlled for smoking, which may be more easily done at the county level using smoking rates than at the individual level if it is not measured. The reason I wonder about the role of smoking is that I think it may be related to mortality since it is so strongly linked to SES. If individuals living in areas of low level of SES are also either more likely to smoke or are more likely to have neighbors, friends, and family who smoke then you may have one mechanism through which social networks impact mortality.

We believe that these considerations are correct. Nevertheless, in this study, as in other ecological studies on socioeconomic inequality and mortality, it has not been possible to incorporate information on tobacco use.

  1. The results suggest an independent effect for mortality of county-level context, however, the analysis lack individual level data. Individual-level SES effects could be more apparent than aggregate SES effects or at the very least that they would provide independent or confounded impacts on mortality; this should be, however, included in the discussion.

-We appreciate the comments regarding the importance of including data at the individual level. In the first two paragraphs of section 4.2 (Methodological strengths and limitations) we have clarified the aspects related to the ecological character of the study, particularly in relation to the effect of the area of residence.

  1. Please have this paper edited by a proficient, native English speaker or, preferably, by one of the many science editing services now available.

-We have carried out a new revision of the translation of the article.

Minor comments

Line 17. The sentence is incomplete, it should be “many studies have analyzed socioeconomic inequalities and its association with mortality in urban areas”.

-In line 17 the suggested change has been introduced.

Line 22, please clarify “specific rates”.

-In line 23 the word “specific” has been deleted.

Line 24-26, two sentences were repeated stated.

-In lines 25-27 one of the repeated sentences have been deleted

Line 40-44, please clarify the changing level of inequality regards for socioeconomic inequality or inequality mortality.

-In lines 41 and 43 the concept has been clarified

Line 50-53, please revise the sentence

-From line 53 to 56, the sentence has been revised and modified.

Line 57-60, please revise the sentence

-Lines 61 to 64, the sentence has been modified.

Line 78-81, please clarify the aim of the study.

-Lines 83 to 89, the sentence has been modified and the aim has been clarified.

Line 103, please clarify “come”

-Line 111, the word “come” has been changed for “were obtained”.

Line 111-112, please edit the sentence, and clarify “mortality indicators”

-Line 121, mortality indicators have been clarified by adding “rates and relative risks”.

Line 115, “specific mortality”. The present study only shown rates and RR for overall mortality.

-Line 125, the word “specific” has been deleted.

Line 171 and Line 177, Figure legend: “specific mortality rates”, it should be over mortality rates by age, gender, country of birth, and DL.

- Lines 180 and 188, the word “specific” has been deleted.

Line 202, please clarify “DL5.ble 5.

-Lines 211 to 212 the repeated sentence has been deleted.

Line 202-203, the sentence is not fully stated.

-Lines 211 to 212, the sentence has been deleted as it was a repetition of the table 5 title.

Line 343, please clarify the inequality regards for socioeconomic inequality or inequality mortality.

-Some changes have been made in line 351 to clarify the concept.